Investigation of time series-based genre popularity features for box office success prediction

Shahid Muzammil Hussain
Islam Muhammad Arshad arshad.islam@nu.edu.pk
Computer Science, National University of Computer and Emerging Sciences , Islamabad , Pakistan
Asif Muhammad
Electronic publication date: 2023 Nov 9
Publication date: 2023
Volume: 9
Electronic Location ID: e1603
Received 2023 Mar 24; Accepted 2023 Aug 29
Copyright: ©2023 Shahid and Islam
Copyright year: 2023
Copyright holder: Shahid and Islam
License: This is an open access article distributed under the terms of the Creative Commons Attribution License, which permits unrestricted use, distribution, reproduction and adaptation in any medium and for any purpose provided that it is properly attributed. For attribution, the original author(s), title, publication source (PeerJ Computer Science) and either DOI or URL of the article must be cited.
License URL: https://creativecommons.org/licenses/by/4.0/

Keywords: Movie profitability prediction, Time series forecasting, Feature engineering, Pre-production phase

Funding: The authors received no funding for this work.

==============================
Predicting the profitability of movies at the early phase of production can be helpful to support the decision to invest in movies however, due to the limited information at this stage it is a challenging task to predict the movie’s profitability. This study proposes genre popularity features using time series prediction. We argue that a movie can produce better box office returns if its genre’s popularity is high at the time of release. The novel genre popularity features are proposed in terms of budget, revenue, frequency, success, and return on investment (ROI). The proposed features couple the predicted genre popularity with release time, in order to train the machine learning classifiers. The experimentation shows that the Gradient Boosting classifier gained a significant improvement using proposed features and achieved an accuracy of more than 92.4%, i.e., 35.7% better than an existing state of the art study considering a multi-class problem.

Introduction

Securing investment is a crucial factor in the production of a movie and investors want to make informed decisions that can lead to having maximum return on investment (ROI). It is desirable for the investors to have the information about success prospects of the movie in pre-production phase. Every movie goes through different phases prior to its release. In literature, movie making process is divided into phases (Lash & Zhao, 2016; Zhao, Xiong & Jin, 2022; Shahid, Islam & Beg, 2023), i.e., development and pre-production phase, production phase, post-production phase, pre-release and released phase. In the development and pre-production phase, the script is developed, casting is decided, budget is estimated, and investment is secured. After acquiring investment, a movie goes into the actual production phase in which movie is shot. After the shooting is completed, a movie enters into post-production phase in which editing is performed and trailer are developed. In the pre-release phase, the focus is on advertising and movies are promoted using social media, print and electronic media to create a movie awareness. In the end, a movie is released to appropriate cinema houses resulting in the profit for investors. Movie box office return (MBOR) prediction during the phases has their respective significance. In pre-production, MBOR prediction is performed with investor’s standpoint while distributor’s standpoint is considered in post-production. Exhibitor’s standpoint is considered in pre-release, and analyst standpoint is considered in post-released phase (Lash & Zhao, 2016).

Portions of this text were previously published as part of a preprint (Shahid & Islam, 2021). The research challenge in predicting the profitability of the movie in the pre-production phase is significant due to the limited amount of available information. In the pre-production phase, available information are related to the movie script, genre, cast and crew, budget and expected release time. This article aims to predict the MBOR to identify that a movie will be hit, average or flop in the box office with the limited information available in pre-production phase. Predicting MBOR is an active research problem and it provides investors with the key information related to the movie profitability. Many of the existing MBOR prediction techniques focus on post-production (McKenzie, Rossiter & Shin, 2020; Ahmad, Bakar & Yaakub, 2020), pre-release (Kim, Hong & Kang, 2015; Choudhery & Leung, 2017) and released phase (Kim, Hong & Kang, 2015; Mundra et al., 2019; Sachdev et al., 2018). These studies are not very useful for investors because the time to make investment related decision has passed. Lash & Zhao (2016) and Lash et al. (2015) have investigated the effectiveness of the features for supporting movie investor assurance system and predicted the movie profitability during pre-production phase. Gao et al. (2019), Ahmed, Waqas & Afzal (2020), Kim, Lee & Cheong (2019), and Sahu et al. (2023) also contributed in MBOR prediction in pre-production phase.

Figure 1 shows that four genre, i.e., action, adventure, comedy, and drama are prominent considering the number of movies as well the revenue generated. The annual variation in the number of movies belonging to each genre and corresponding volatility in the revenues generated that represent the risks involved in movie production business. Therefore, the importance of timely prediction for the movie profitability increases manifold. Figure 1A highlights that there is a higher number of movies belonging to the drama genre. The budget of a drama movie is usually lower than that of an action or adventure movie (Johnsen, 2023). Additionally, movies belonging to the drama genre tend to attract audience easily (Gao et al., 2019). The revenue of a drama genre movies does not correspond with the number of drama genre movies. The number of movies belonging to comedy genre appear to be consistent through out the time period. Moreover, the number of movies belonging to horror and thriller/suspense genre movies and the revenue of the corresponding movies has increased in recent years. Figure 1B shows that action and adventure movies are dominating the box office and have a earned higher revenues. Additionally, action and adventure genre consume larger budgets (Leung & Qi, 2022) therefore these genres have low share related to the number of releases as seasoned investors consider these movies for investment (McKenzie, Rossiter & Shin, 2020).

Figure 1 Genre based Hollywood movie industry analysis (1990–2018, 4,031 movies).

Research gap

In the existing literature, various movie attributes have been used to make predictions while the genre popularity trend and its impact on profitability is not considered in making box office success predictions. We argue that popularity of a particular genre at the time of the movie release plays a significant role in the success of the movie. The popularity of a genre may vary depending upon various social factors. Therefore, release time of movie has an impact the success of the movie have not been examined in the literature as well. Genre popularity can be predicted using time series analysis to engineer novel features that can play vital role in accurate MBOR prediction.

Portions of this text were previously published as part of a preprint (Shahid & Islam, 2021). To substantiate the research gap, we have extracted genre popularity trends from the “The Movie dataset” (Banik, 2017). Figure 2A shows the revenue based behaviour of the top 5 genres for the specific time of the year, i.e., month of January for the last 10 years. It can be observed that no particular genre remains on top, and genre position varies every year that shows absence of a periodic pattern. Action and horror genres have been more profitable in 2011 where as thriller genre has taken the top position in 2014. In Fig. 2B, genre analysis of the last 3 year is presented to show genre variation in each month. It can be observed that the genre does not show the regular trends. Moreover, these plots become the basic motivation of proposing new time series based features. Additionally, such variations are also observed by Perno (2022) showing genre popularity variations in the last 80 years. The analysis shows that musical genre used to be popular in 1940's and the comic genre is the most favorite genre in the last 20 years.

Figure 2 Revenue based genre variations in different time periods.

Problem statement and research questions

The existing works use movie metadata for predicting movie success in the pre-production phase (Lash & Zhao, 2016). The existing studies have used the data for the movie attributes but have not considered the temporal variation of popularity of movie’s genre. As discussed earlier, features representing genre popularity are transformed into time series (TS) (Mun & Chong, 2018; Redfern, 2012; Ru et al., 2018) and predicted genre popularity metrics are used to enhance movie box office success prediction accuracy.

This research addresses the following research questions:

• RQ 1: How accurately genre popularity can be predicted?

Explanation: This research defines genre popularity feature in five dimensions i.e., revenue, budget, frequency, success, and ROI. Based on these aspects, multiple monthly time series are created using weekly movie box revenue dataset, then the SARIMAX model (statsmodels, 2020) is trained on the dataset to forecast the genre’s popularity.

• RQ 2: How accurately can we predict the success of a movie by including the new features?

Explanation: To predict the movie profitability in the pre-production phase, the feature set and results of the existing best performing works are re-produced. Using existing pre-production based features, and newly proposed genre popularity features are merged and multiple ML classifiers are evaluated with respect to their capability to predict the success of a movies.

• RQ 3: How the ML models react with respect to box-office success when the release time of the movie is altered?

Explanation: We argue that temporal nature of genre popularity influences movie profitability. This is verified by altering movie release time to observe the change in behavior of ROI based movie profitability class. When genre popularity features value changes, it changes the ROI based class, which shows genre popularity based release time does influence the movie profitability.

Contribution

This research focuses on forecasting genre popularity using multiple univariate-time series, primarily based on genre’s popularity features to improve the movie profitability prediction results and also investigate the effects of change in the release time. The aim is to investigate time series based novel features that promise to provide investors relatively accurate predictions about the box office performance of movies in the early stages, that is the time before investing in any movie. The contributions of this work in the movie profitability prediction problem are following:

• Proposed time series based genre popularity features measure in terms of revenue, budget, success, frequency and ROI.

• Improved the accuracy of movie profitability prediction by incorporating genre popularity features considering the multi-class problem.

• Conducted the thorough investigation of each genre’s popularity related to release time.

The remaining article is organized as follow. ‘Related work’ provides a review of the literature and comparison analysis of movie box office revenue and time series. ‘Methodology’ presents the research methodology in which dataset is described, data exploratory analysis is performed, and feature engineering are discussed. ‘Experimental Details’ contains the discussions related to evaluations and experiments performed. ‘Results and Discussion’ discusses the results and the article end with the concluding remarks in ‘Conclusion and Future Work’. The pre-print version of this work is available at Shahid & Islam (2021).

Related work

Movie box office revenue prediction

Portions of this text were previously published as part of a preprint (Shahid & Islam, 2021). Limited work in literature is found related to the pre-production phase prediction because limited information related to movies is available in pre-production phase. Sahu et al. (2023) focuses on the Indian movie industry, highlighting its combination of several movie industries and its position as the largest producer of films globally. It utilizes the IMDb dataset and proposes a predictive model for early-stage movie success, achieving 96% accuracy. The study aims to establish success criteria specific to Indian movies and plans to consider age-wise ratings and classify movie popularity in future work.

Lash & Zhao (2016) has worked on predicting movie profitability in an early stage of the movie to support movie investment assurance. The authors have proposed the movie investment assurance system (MIAS) and used the pre-production phase features like genre, movie plot, budget, and information related to movie actors and director. Based on these basic features, they performed the extensive features engineering and extracted nearly 48 features that are divided into four groups, i.e., Who, When, What and Hybrid features group. Additionally, they have engineered features using collaboration network analysis, topic modeling and statistical analysis. In Lash et al. (2015) and Razeen et al. (2021), authors have predicted the movie profitability at early stages and have used a linear regression model to predict the movie profitability. In Lash & Zhao (2016) dataset contains 2,506 movies from 2000 to 2010 while in Lash et al. (2015), 1,353 movies are considered in the dataset. In Lash & Zhao (2016), authors have explored new features but did not studied the genre trends and popularity to predict movie success. Additionally, revenue-based temporal genre features are defined considering 1 year time window (cumulative revenue of all movies belonging to a selected genre released in a year) however, this research defines time series based genre features comprising of five different aspects of the movies evaluated on weekly basis. Moreover, 90.4% accuracy is achieved when the movie profitability is defined as binary classification problem in Lash & Zhao (2016) while 73% accuracy is achieved in case of multi-class classification problem. This research considered movie profitability as multi-class classification problem, therefore we claim that our experiments have produced better results while considering a multi-class movie profitability problem.

A successful movie will be the one that has success on box office and also have received good reviews (Gao et al., 2019), however it can be argued that in exceptional cases a movie may get negative reviews and may still be able to earn reasonable box office performance. For example, a movie, “Fifty Shades of Grey” has earned the profit of $520 million but it got 4.1 IMDB rating. Gao et al. (2019) uses most of the features define in Lash & Zhao (2016) to perform the binary classification and achieve 79.15% accuracy. Gao et al. (2019) used the Movie rating in class definition, while movie rating is not a pre-production based feature. Also, the temporal aspect of the movie features are also not explored.

Ahmed, Waqas & Afzal (2020) investigated on the movie success prediction in the pre-production phase and using the hybrid approach, which is voting based classification; using Random forest, SVM, Gradient boosting and XGBoost model to predict the movie success. 18 features such as genre score, director and first three actor’s rating, experience, score and Facebook page likes are used for training and achieved the accuracy of 85% on multi classification. Similarly, authors in Ni et al. (2022) authors employed various machine learning algorithms, including XGBoost, light gradient boosting machine (LightGBM), categorical boosting (CatBoost), gradient boosting decision tree (GBDT), random forest (RF), and support vector regression (SVR), to construct a box office prediction model. Additionally, a second layer meta-learner was introduced, which utilized a multiple linear regression model. In (Ahmed, Waqas & Afzal, 2020) Facebook likes as a feature is used, which cannot be considered as a pre-production based feature.

Portions of this text were previously published as part of a preprint (Shahid & Islam, 2021). In the pre-production phase movie revenue prediction, Zhang et al. (2015), Sahu et al. (2023) also contributed by constructing a model to forecast the movie box office revenue. Hunter & Smith (2016) also worked on the pre-production phase, and predicted the opening weekend box office of 170 USA based movies from 2010 to 2011 with 0.51 MSE. Mundra et al. (2019) have used social media to predict the movie revenue in released phase using the analysis of movies related tweets to predict movie popularity. They built a prediction model using tweets sentimental analysis and IMDB dataset. Similarly another study focuses on predicting movie success based on rating and temporal popularity using tweets (Alhijawi & Awajan, 2022). The proposed temporal product popularity model (T-PPM), aims to forecast the movie’s temporal popularity using a random forest classifier using a dataset called TweetAMovie collected from IMDb and Twitter.

In movie pre-released phase, information related to movie buzz, star buzz and presence on social media plays a significant role. Movies have a larger presence on social media has high chances of getting box office hits. Social media contains information related to user preferences and likelihood. Choudhery & Leung (2017) have predicted the movie success by focusing on tweets mining and using them for box office success prediction.

To provide prediction of a movie’s performance at an earlier stage, Kim, Lee & Cheong (2019) proposed a deep-learning based approach to predict the success of a movie using only movie plot summary. Kim extracted sentiments (positive, negative, neutral and compound) from movie story and created ELMO embedding of movie script. Wang et al. (2020) investigated the influential factors and propose a dynamic heterogeneous network embedding model to learn movie participants and predict the movie box office revenue.

Time series forecasting

Mun & Chong (2018) have investigated the daily box office sales forecasting and have used total and split exponential smoothing algorithm. Three forecasting models are developed to predict the movie revenue in Kim, Hong & Kang (2015). Movie revenue of prior, a week after and two weeks after release are forecasted using social network service and machine learning techniques. In Chen et al. (2019), authors perform the time series analysis of tourism demand. Time series (TS) is divided into quarterly time series and split into 4 groups, and each group represents the tourist demand in that quarter. Multi-Structured Time Series Model (M_STSM) has been used and achieve a mean absolute percentage error score of 9.46%.

Ru et al. (2018) work on the box office time series and predicted the daily box office using end-to-end deep learning model (Deep-DBP) by considering the static and temporal features to forecast the daily box office sales with the prediction error of 30.1%. In the time series forecasting, the ARIMA is one of the most widely used approaches (Hyndman & Athanasopoulos, 2018). Blázquez-García et al. (2020) presented the energy consumption module based on the Seasonal ARIMA (SARIMA) statistical model to forecast the energy consumption of a green-elevator. SARIMA is an extension of ARIMA that explicitly supports uni-variate time series data with a seasonal component (Hyndman & Athanasopoulos, 2018).

For the operational solar forecasting for the real-time market, Yang, Wu & Kleissl (2019) uses SARIMA model for multi-step ahead time series forecasting and generated 25-step-ahead forecasts using 15-min ground data. McHugh et al. (2019) focuses on electricity price forecasting through time-series application with real energy data and used the Seasonal Auto-Regressive Integrated Moving Average model with eXogenous variables (SARIMAX) as electricity prices follow a seasonal pattern controlled by various external factors for achieving energy efficiency (Shahid et al., 2020). Khatibi et al. (2020) uses SARIMAX for predicting tourism demand by exploiting Social Media and Environmental features to perform accurate fine-grained predictions.

Limited research work found related to movie profitability prediction in pre-production phase is shown in Table 1. The existing work in the pre-production phase has used movie genre as a categorical feature and did not take the benefit of this important feature. It is noteworthy that none of the existing literature considers genre popularity as time-series. The three references identified in bold are used for comparison analysis in ‘Results and discussion’. The aspect of variation in genre popularity that may translate into profitability is also not examined in any of the existing research. Movie dataset used in previous work also have number of movies and the prediction efficiency of the techniques used is low in their work due to lack of effective features and models.

Table 1 Comparison matrix of movie box office profitability prediction.

Ref.	Contribution	Technique	Results	Features	Prediction Phase	Dataset	
Lash et al. (2015)	Predicted the movie profitability at early stages to increase investor certainty.	Logistic regression	Accuracy 77.1%	Average annual profit, APPG, AWPG, Release dates, Team heterogeneity, Average degree, betweenness centrality, tenure, actor gross	Pre-production	test	
Ghiassi, Lio & Moon (2015)	Dynamic Artificial Neural Network (DANN) for the forecasting of movie revenues in the pre-production period.	DANN	Accuracy = 94.1%	MPAA rating, pre-release advertising expenditures, screen count, production budget run time, sequel, seasonality	Post production	test	
Zhang et al. (2015)	Uses movie fans as a quantization of director and star value and forecasted movie revenue.	CART	Accuracy (APHR) = 76%	Production Country, Genre, Seasonality, Star value	Pre-production	test	
Hunter & Smith (2016)	Examined the movie plot textual properties to predict the movie opening weekend gross.	OLS	MSE = 0.51	Box office revenue, size of the text network	Pre-production	Multiple Sources for Movie Scripts	
Lash & Zhao (2016)	Predicted movie profitability to support movie investment decisions at an early phase.	Random Forest	Accuracy = 73%	Basic Features: Genre, Plot Synopsis, Budget, Revenue, Team(Actor, Director)	Pre-production	IMDb and Box Office Mojo	
Choudhery & Leung (2017)	Uses tweets and their sentiments to predict box office revenue of the movie.	Polynomial regression model	MSE = 13%	Tweet counts, positive tweets %, negative tweets %, age demographic, sentiment, public reception.	Pre-release, Released	Tweet Dataset	
Sachdev et al. (2018)	Effectively estimate the box-office gross revenue for a movie using the public information available after its first weekend of release.	Linear Regression and DTR	MAPE: 24.76%.	Title, genre, release date, budget, No. of screens in OW, OW revenue, IMDB rating, TomatoMeter, TomatoRating, IMDB Popularity, Rotten Tomatoes, and domestic revenue	Released	MDB and Rotten Tomatoes	
Mundra et al. (2019)	Performs movie related Tweets sentiment analysis to predict movie success.	Random Forest	Accuracy = 93.17%	Revenue, Budget, Actor 1, 2 and 3, movie Genres, Director, Duration of the movie, No. of users voted, Content rating, Title year, PT/NT ratio	Released	TWeet Dataset	
Gao et al. (2019)	Analyze the movie success based on critical and financial perspectives.	SVM	Accuracy = 79.15%	Basic Features: Genre, Plot Synopsis, Budget, Revenue, Team(Actor, Director) and movie Rating	Pre-production	IMDB	
Ahmed, Waqas & Afzal (2020)	Forecast box-office success, in the early stage using voting technique	SVM, GdB, XgBoost, RF	Accuracy = 85%	Genre, Director and 3-Actor Rating, Experience,score and FB likes.	Pre-production	Movie Trailer Reviews from Youtube	
Kim, Lee & Cheong (2019)	Proposed a deep learning approach using the ELMO embedding and movie sentiment	ELMO, a merged 1D CNN, residual LSTM	F1: 0.68, 0.70	Plot summary text, Genre	Pre-production	CMU Movie Summary Corpus	
Ahmad, Bakar & Yaakub (2020)	Using reviews on movie trailers on YouTube, moviegoers intentions to purchase tickets can be predicted.	Multi Linear Regression	Relative AE = 29.65%	Budget, ReviewsCount, ViewsCount, WPNratio, Genre, Rating, LDratio, Duration, PI	Post production	Naver Movies	
Zhao, Xiong & Jin (2022)	Movie sales prediction with by evaluating the influence of microblogs.	LR, SVR	Relative AE = 29.65%	Microblogs, Likes, Comments, Forwards	Post production	China Box Office Dataset	

Methodology

Portions of this text were previously published as part of a preprint (Shahid & Islam, 2021). Dataset containing the information for time series generation has been collected using the scrapping online resource “The-numbers” (The Numbers, 2020) and it is supplemented with an existing presented in Banik (2017) using the steps described in the ‘Pre-Processing’ section. After pre-processing, raw features are extracted to engineer novel features that are used to improve the prediction efficiency of the movie box office success in the pre-production phase. The methodology followed for the conducted experiments in this work is presented in the Fig. 3, that shows the workflow of the experiments and task performed.

Figure 3 Methodology used for predicting MBOR in the pre-production phase using time series analysis.

Dataset Collection

The Movie dataset (Banik, 2017) contains 45,466 movies from 1900 to 2018. This dataset includes the information related to movie metadata, cast and crew of the movies. Python library Beautiful Soap (Beautiful Soup, 2020) is used to crawl the the-number website (The Numbers, 2020) to scrap the weekly revenue data containing the following attributes: Date, Rank, Gross, % Change, Theaters, Per Theater, Total Gross, Week. As referred by the data source, Gross means revenue generated in that particular week while Total Gross means revenue generated from release date to current week.

Table 2 shows the list of existing features that are presented in the literature whereas MIAS_TS (Movie Investment Assurance System using Time Series) shows the features used in our work. These features may be broadly classified into rating features (scores of actors, directors, movies) on social media and monetary features (profitability or revenue of actor, directors, movies). Cast ADC (Actor-Director Collaboration) Freq and Cast ADC Profit refer to the number of movies an actor and the director worked together and the profit generated by the corresponding movies respectively. AGE (Average Genre Expertise) of an actor refers to the number of movies belonging to a particular genre actor has performed.

Table 2 Feature set used in the experiments.

No	Feature name	Lash & Zhao (2016)	Gao et al. (2019)	Ahmed, Waqas & Afzal (2020)	MIAS_TS	No	Feature name	Lash & Zhao (2016)	Gao et al. (2019)	Ahmed, Waqas & Afzal (2020)	MIAS_TS	
1	Dir Avg Gross	✓	✓		✓	2	Budget Cont				✓	
3	Cast ADC Profit	✓	✓		✓	4	Genre Revenue Share				✓	
5	Dir Total Gross	✓	✓	✓	✓	6	Cast Avg Movies				✓	
7	Genre Budget Share				✓	8	Avg Annual Profit	✓			✓	
9	Cast ADC Freq	✓	✓		✓	10	Cast Total Rev	✓	✓		✓	
11	Cast top AvgProfit		✓		✓	12	Cast Novelty	✓			✓	
13	Cast Top Profit	✓	✓		✓	14	Cast Total Movies				✓	
15	Genre Freq share				✓	16	Total Cast Count				✓	
17	Genre ROI Share				✓	18	AGE	✓			✓	
19	Genre Success Share				✓	20	Total Tenure	✓	✓		✓	
21	Season	✓	✓		✓	22	Competition	✓			✓	
23	Genre (avg)	✓	✓		✓	24	Plot Analysis	✓	✓		✓	
25	Budget		✓			26	Year		✓			
27	Dir Avg Rating		✓			28	Cast Avg Rating		✓			
29	ADC Rating		✓			30	WAGE	✓				
29	Dir Total Profit	✓				30	Dir Avg Profit	✓				
31	Cast Mean Rev	✓	✓			32	Cast Mean AvgRev	✓	✓			
33	Average Tenure	✓	✓			34	Cast Total Profit	✓	✓			
35	Cast total FB Likes			✓		36	Cast Avg Profit		✓			
37	Day		✓			38	APPG	✓				
39	AWPG	✓				40	GenreScore			✓		
41	Actor 1 Experience			✓		42	Actor 1 Score			✓		
43	Actor 2 Experience			✓		44	Actor 2 Score			✓		
45	Actor 3 Experience			✓		46	Actor 3 Score			✓		
47	Actor 1 Rating			✓		48	Actor 1 FB Likes			✓		
49	Actor 2 Rating			✓		50	Actor 2 FB Likes			✓		
51	Actor 3 Rating			✓		51	Actor 3 FB Likes			✓		
52	Dir Rating			✓		53	Dir FB Likes			✓		
54	Dir Experience			✓		55	Cast total Avg Profit	✓	✓			

Pre-processing

Portions of this text were previously published as part of a preprint (Shahid & Islam, 2021). The Movie dataset (Banik, 2017) contains more than 30 features however we have selected eight features that are available during pre-production phase, i.e., budget, genres, overview, revenue, release_date, cast, crew and spoken_languages. Overview feature contains movie summaries, which is used to extract topic modeling features and to perform movie plot analysis. These features are further used to engineer novel features for model training as discussed Section ‘Feature Engineering’.

Budget information is the key feature that is used to determine the ROI value however, this information is not available for several movies in the Movie dataset and this aspect has reduced the dataset size. Movies having English as their language are selected using basic feature “spoken_languages” and movies with missing values for above mentioned 8 features have been removed resulting in a dataset of 5019 movies as shown in Fig. 3. Table 3 shows the distribution of movies in different genres and classes. Weekly revenue data of these 5019 movies is scrapped from The Numbers (2020) for creating time series based features. All of the revenue values are expressed in US dollars.

Table 3 Movie distribution in each class label.

Class	Adventure	Action	Western	Comedy	Drama	Thriller	Horror	Romance	Music	Documentary	DS: 70/30 (Tr)	
Hit	262	232	7	319	356	136	136	28	2	18	1,097	
Average	336	413	18	423	430	204	113	50	9	11	1,385	
Flop	183	284	6	288	453	162	71	32	11	16	1,059	
Class	DS:1990-95		DS:1995-2000		DS:2000-05		DS:2005-10		DS:2010-15		DS:70/30	
	TR	TE	TR	TE	TR	TE	TR	TE	TR	TE	TE	
Hit	486	107	593	119	712	171	883	214	1,097	271	409	
Average	312	121	433	224	657	307	964	421	1,385	431	622	
Flop	190	128	318	198	516	224	740	319	1,059	271	447	
Notes.

DS Dataset

TR Training set

TE Test set

Exploratory data analysis

Dependency of Genre on budget and revenue.

To support the hypothesis that every genre has different revenue, budget, frequency, success and ROI trends, plots are presented using the weekly revenue data. Genre based trend related to revenue are also plotted because to analyze the revenue generation capability of each genre.

Based on Fig. 4A, it can be observed that each genre has different trends with respect to the ROI. In the context of ROI, horror genre has dominance which make it more suitable for investment from investors standpoint. Based on Fig. 4B, horror and documentary genre has positive ROI in their first week, which shows that investors can have their investment recover in their first week. However with as the time progresses, weekly ROI decreases. The aim plotting weekly ROI is to show the diversity in the profitability of genre. Some genre are able to recover the investment within first week while rest may take longer screen time duration. Figures 4C and 4D show that most successful genres with respect to revenue are action and adventure. It also shows that successful genres based on ROI (Fig. 4) are different than that of revenue based.

Figure 4 Genre based ROI (A, B) and revenue (C, D) trends.

Portions of this text were previously published as part of a preprint (Shahid & Islam, 2021). It is noteworthy that investors do invest in the low ROI yielding ROI and there can be various reasons. Certain movie genres, while having a lower overall ROI, may have a passionate and loyal fan base. Investors may see the potential in catering to this specialised demographic and believe that the film can still turn a profit (Johnsen, 2023; McKenzie, Rossiter & Shin, 2020). Additionally, some film genres may not provide high financial returns, but they may gain critical praise and industry reputation. Investors may be motivated by a desire to promote original and thought-provoking films that add to the artistic landscape (Lauria & Phillips, 2021).

In Fig. 4D, weekly revenue of the movies is presented. It shows that the movie generates most of the revenue in their first month of release. On average 23% of the movie revenue is generated in their first week of release while in the first 4–5 weeks of release, a movie generates 55% of its total revenue, 8% in next 4 week and this percentage decreases further as time progresses.

Number of weeks, a movie occupies cinema screens plays a critical role in the profitability of the movie. Figure 5A shows mean number of weeks of screen occupation by movies belonging to particular genre. It shows that movies belonging to drama genre are produced the most however their screen drop rate is also higher as compared to movies belonging to other genres. Action and adventure movies are produced less in number and their screen drop rate is not as not as high as drama and comedy movies. The reason for popularity of drama movies include, wide audience appeal, low production cost, cultural reflection, and potential for repeat viewing (Manuel, 2023).

Figure 5 Genre based movie count and budget distribution.

Movies belonging to drama and comedy genres account for the bulk of the production pie followed by Action and Adventure, the later two account for the majority of the budget as shown in Fig. 5B. The reason for popularity of drama and comedy banks primarily on the aspect of story telling. We can also observe that the budget distribution for different genres that adventure and action genres movies have more budget than others. Higher budget is correlated to higher revenue and not necessarily correlated to higher ROI or profit (McKenzie, 2023). As shown in Fig. 5B, horror genre movies have a low budget, however horror movies have a relatively higher ROI.

Feature engineering

In the pre-production phase, basic information related to movies is available, e.g., movie plot, cast and crew, budget,and genre. Therefore, novel features must be engineered to increase the effectiveness of available information to achieve more accurate predictions. In existing works, Lash & Zhao (2016), Gao et al. (2019) have performed the feature engineering using basic features as well as social network based features. However, this work proposes the use of composite features that are extracted using time series.

Portions of this text were previously published as part of a preprint (Shahid & Islam, 2021). This section provides the description of the novel features based on time series. Considering the set of movies M, belonging to genre g ∈ G with the actors and cast represented by set A and C, we have modelled the time-series’ using the following variables.

Movies = M = m1,m2,⋅⋅⋅mn

Time Period = T = t1,t2,⋅⋅⋅tj

Genres = G = g1,g2,⋅⋅⋅gx

Time series based feature

• Genre Freq Share: This time-series is formulated to analyze the trend of certain genre based on a number of movies. It represents the ratio of number of movies released of certain genre to number of movies movies released in a time period ti. (1) Genre_Freq_Shareti,gk=NgktiNti

Ngkti is the total number of movies belonging to genre gk in time period ti and Nti represent the movies released in ti.

• Genre Revenue Share: This time-series is formulated to analyze the trend of revenue of movies belonging to certain genre. It represents the ratio of revenue of movies of certain genre to the revenue generated by movies in time period ti. (2) Genre_Rev_Shareti,gk=∑Ngktiμgk∑Ntiμ

μgk and µrepresent the revenue of movies belonging to genre k and all movies respectively.

• Genre Budget Share: This time-series is formulated to analyze the trend of budget in a certain genre movie. It shows, how much investment or budget is invested in certain genre in Time period ti in comparison to the overall budget of other movies in that time period. (3) Genre_Budg_Shareti,gk=∑Ngktiβgk∑Ntiβ

βgk and β represent the budget of movies belonging to genre k and all movies respectively.

• Genre ROI Share: This time-series is formulated to analyze the trend of ROI in a certain genre movie. It will show how much ROI, movies of certain genre has earned in time period ti in comparison to the overall ROI generated in that time period on average. This feature represents the strength of genre over a period of time in movie industry. (4) ROIi=μti−βtiβti

μti and βti represent the revenue and budget of movies that are being released in time period ti respectively. (5) Genre_ROI_Shareti,gk=∑Ngktiγgk∑Ntiγ

• Genre Success Share: This features is adapted from Redfern (2012) and is used to find the success ratio of a certain genre in a certain time period over other genre movies. In Redfern (2012) success is defined as ROI > 10%, Redfern (2012) have used top 50 movies of the year to study the success trend of the genre. This study defines success share of a genre gk in time period ti using Eq. (6). (6) Genre_Success_Shareti,gk=MgktiΓ

Mgkti represents the number successful movies belonging to genre k in time period i and Γ represent the total successful movies in time period i. A successful movies is defined using the ROI, i.e., any movie having ROI > α is considered successful (Gao et al., 2019) where α = 1

Portions of this text were previously published as part of a preprint (Shahid & Islam, 2021). Algorithm 1 is used for preparing time series data to represent genre popularity in term of ROI. To prepare the time series data, mean ROI is calculated to see the performance of movie industry in a certain time period. Then mean ROI is calculated to see the performance of movies belong to certain genre g in a certain time period t. Other time series i.e., Genre’s Revenue Share, Freq Share, Success Share, Budget Share are also generated in a similar manner.

_______________________ Algorithm 1 Time Series Computation ComputeTS(Movie_Data)_______________     Input: Movie revenue data metric with columns: Date, Gross, ROI, Genre     Output: List of Time series with ”Genre ROI Share” values of all Genres   1:  Key ← Extract_ReleaseDate(Movie_Data)   2:  Movie_ROI ← Movie_Data[Key].Mean_ROI     {Re-sample the data to Month Start ’MS’ for a month based time series      data }  3:  Movies_ROI_MS ← Movies_ROI.resample(′MS′).sum()   4:  initialize(Genre_TS)   5:  for all  Genre  do  6:     GMovies ← Movie_Data[Genrei]   7:     GMovies_ROI ← GMovies[Key].Mean_ROI  8:     GMovies_ROI_MS ← GMovies_ROI.resample(′MS′).sum()         {Divide the current genre’s monthly ROI value series with the overall         market ROI monthly value series}  9:     GMovies_ROI_TS ← GMovies_ROI_MS/Movies_ROI_MS 10:     append(Genre_TS , GMovies_ROI_TS) 11:  end for 12:  return  Genre_TS______________________________________________________________

A few non-time-series based proposed features have been adapted from the previous studies as given below.

• Cast count: It is defined as total number of actors cast in the movie. It tells us the movie size and diversity. If the cast count is high, then there are chances to attract more audience because of each actor’s star power effect (Lash & Zhao, 2016).

• Cast total movies: It is defined as sum of all movies participated by cast of the selected movie. It is basically a count of the movies, in which the current movie’s cast and crew has appeared in the past. It tells us the overall experience of the movie’s team.

• Budget Contribution: It is a temporal feature that is purposed in this work. The budget contribution of a movie mi with respect to all movies having same genre that have same expected month of release. In Eq. (7), the budget contribution of the genre gk at time period ti and it is derived using Eq. (3). (7) Bd_Cmi=bmi∗Genre_Budg_Shareti,gk.

Time series forecasting

Portions of this text were previously published as part of a preprint (Shahid & Islam, 2021). Using time series features like Freq_share, Revenue_share, ROI_share, Budget_share and Success_share multiple univariate time series of genres are created. Genre popularity is forecasted for the time period related to the release time of the selected movie. Genre popularity features is further used with other features to predict the movie box office success. For forecasting genre popularity, SARIMAX (statsmodels, 2020; Hyndman & Athanasopoulos, 2018; Mills, 2019) and LSTM (Gers, Schmidhuber & Cummins, 1999) models are used however, the results pertaining to SARIMAX models are discussed in detail due to its better performance.

Genre popularity is forecasted in the context of budget share, revenue share, ROI share, frequency share and success share, using time series prediction. The popularity metrics are further used in the feature set for movie revenue prediction. In line 2, SARIMAX model is used for time series forecasting, and it utilizes the standard parameters representing auto regression, differences, moving average, seasonal component, i.e., (p, d, q) = (1, 1, 1), seasonal_order(P, Q, D, s) = (1, 1, 1, 12) obtained using grid search (Hyndman & Athanasopoulos, 2018).

Genre popularity (GP): It is multiple uni-variate time series based feature and it is defined using Eqs. (1)–(6). Genre popularity is expressed in the form of 5 time series. It will tell the market strength and acceptance for certain genre during its release time. If the movie genre belongs to popular genre, then it may be successful.

Experimental Details

Portions of this text were previously published as part of a preprint (Shahid & Islam, 2021). From the dataset, new pre-production phase movie features enlisted in Table 2 are created. These proposed features are then merged with the existing features resulting in 55 features that are further reduced using correlation analysis to 24. The feature data is split into training and test set for classifier training purpose in order to predict the movie box office profitability representing the success of the movie.

Prediction model

Portions of this text were previously published as part of a preprint (Shahid & Islam, 2021). In this work, machine learning models that is trained and tested are random forest (Lash & Zhao, 2016; Mundra et al., 2019; Ho, 1995), support vector machines (SVM) (Lash & Zhao, 2016; Mundra et al., 2019; Gao et al., 2019), boosting (Ahmed, Waqas & Afzal, 2020), XgBoost (Ahmed, Waqas & Afzal, 2020), and multi-layer perceptron (MLP) (Lash & Zhao, 2016; van Gerven & Bohte, 2018). These models are used in literature for predicting movie profitability and they have used multiple machine learning models to evaluate their results and then selected the best performing model, so that why this strategy is adopted.

Performance comparison

To evaluate the performance of the proposed mechanism on multi-class classification, comparison with the existing pre-production phase research is performed, and three recent most research work related to profitability prediction in pre-production phase are selected. Benchmark 1 is based on Lash & Zhao (2016) work and gets 80.4% accuracy on evaluating on the current dataset. Benchmark 2 is based on Gao et al. (2019) and results showed that it achieved 84.1% accuracy. Another recent work, Ahmed, Waqas & Afzal (2020) is also considered as evaluation benchmark 3. These research work (shown in bold in Table 1) are selected as benchmarks because the features used by them belong to the pre-production phase. Additionally the success definitions (class labels) used in the benchmarks are based on ROI. In Table 2, comparison of the features is presented. A total of six data instances are created for the purpose of training and testing is presented in Table 3.

Measure of success: In this work, movie success is defined using Lash & Zhao (2016) criteria which measure movie profitability using ROI. Success is divided into three classes, i.e., Hit, Average and Flop. The movie is considered successful or Hit if its ROI falls in the top 25% of the ROIs, Flop if its ROI falls in the bottom 25% and the Average class is assigned if its ROI is between top 25% and bottom 30%. These boundaries threshold translates to ROI ≥ 2.5567 for the Hit class label, ROI ≤ 0.0049 for the Flop class label and if the movie ROI is in range of 0.0049 <ROI ≤ 2.5567 then Average class label is assigned. Table 4 shows the class label and its ROI range (prescribed in Lash & Zhao (2016)) along with the number of movies in that class. The details of the code used to obtain the results is available at Shahid & Islam (2020).

Results and Discussion

Portions of this text were previously published as part of a preprint (Shahid & Islam, 2021).

Genre popularity prediction (RQ1): The first research question is investigated using the time series analysis of genre-based features using SARIMAX. Monthly time series’ are created for the genre popularity and the time series has 564 data points (47 years x 12), where x-axis represents time in months (Jan,1972–Dec,2018) and the y-axis represents the proposed time series based features, i.e, “budget share”, “revenue share”, “ROI share”, “frequency share” and “success share”. After training the SARIMAX model on time series dataset belonging to time-period 1972–2018, next 9 years (2010–2019) are used for genre popularity forecasting. The forecasted genre popularity is then used as features for the movie success prediction.

Portions of this text were previously published as part of a preprint (Shahid & Islam, 2021). Figures 6 and 7 show the mean square error (MSE) and forecasted time series for gross share (adventure genre) and budget share (action genre) respectively for the time period 2010–2018. These graphs show low MSE showing the effectiveness of SARIMAX time series forecasting model.

Table 4 ROI based movie classification criteria and thresholds.

Class	Criteria ROI range	No. of movies	
Hit	ROI in top 25% ≥ 2.5567	1,506	
Average	ROI in between top 25% and bottom 25% 2.5567 > RO I > 0.0049	2,007	
Flop	ROI in bottom 25% <0.0049	1,506	

Figure 6 SARIMAX performance for gross share TS.

Figure 7 SARIMAX performance for budget share TS.

These MSE plots also show that genres behave differently in each definitions like MSE of action and adventure genre is high in “budget share” and “gross share” while the drama genre has high MSE in term of “success share” and “frequency share”. It shows that dominating genre’s forecasting is bit difficult as action and adventure are dominant in terms of revenue (Fig. 4C) and budget (Fig. 5B), while in terms of frequency (Fig. 5A) and ROI (Fig. 4A), drama and horror genre are dominant respectively.

Another interesting observed aspect in the MSE error plots is that MSE is not gradually increasing with the increase in the forecast horizon. It shows that seasonality property maintains somewhat static behavior over the period of selected 10 years.

Portions of this text were previously published as part of a preprint (Shahid & Islam, 2021). In order to evaluate whether genre popularity follows a regular pattern, following experiment is designed to observe the difference between forecasted values and synthetically generated time series. Synthetic time series are generated for comparison with the SARIMAX model’s forecasted value to study the behaviors of genre popularity time series. Following time series’ are interpolated by replacing the “real” time series values with the synthetic values in the following way:

• Last 10 year: Last 10 year TS values (2000–2008) are replaced with the TS values (2010–2018)

• Last year: Current year values are replaced with last year, i.e., TS[t] =TS[t-1]

• 2009: Replaced the time series values (2010–2018) with data belonging to year 2009.

• 2008: Replaced the time series values (2010–2018) with data belonging to year 2008.

Root mean square error (RMSE) of these synthetic time series of the adventure genre measures in “Freq Share” is presented in Fig. 8. It is observed that “SARIMAX-Forecast” is close to the “real” time series pattern while other synthetic time series’ are not close it. This shows that the SARIMAX model is not simply replicating the periodicity and has learned the patterns effectively.

Figure 8 Behavior of adventure genre’s “Freq share” TS.

For first forecast horizon, RMSE of 0.021 is observed for SARIMAX model whereas 0.050 is observed for synthetic “2008” series. Moreover RMSE of 0.045 is observed for synthetic “2009” time series, RMSE of 0.04 is observed for synthetic “Last year” time series whereas RMSE of 0.042 is observed for synthetic “Last 10 year” time series. It highlights that model performance is better than synthetic TS values.

Portions of this text were previously published as part of a preprint (Shahid & Islam, 2021). Based on the comparison between SARIMAX model forecasted time series and synthetic time series’, on average SARIMAX model has 85.76% less RMSE error than synthetic time series. LSTM model is also trained on the genre popularity time series dataset, but RMSE of LSTM forecasted values is higher than SARIMAX forecasted values.

Movie profitability prediction (RQ 2): After forecasting the genre’s popularity in terms of “Budget share”, “Revenue share”, “ROI share”, “Frequency share” and “Success share”, the next phase is to evaluate the effectiveness of the proposed features to predict the movie profitability. As mentioned above, this study has supplemented the novel genre popularity features with the existing features defined in recent studies. Pairwise Pearson correlation (Benesty et al., 2009) is used for feature reduction. After omitting the redundant features, 16 previously existing features are selected and coupled with novel 5-time series based genre popularity features along with 3 non-time series features discussed in Section ‘Feature Engineering’. The complete list of selected features used in the experiments along with the features importance is presented in Table 5 and final feature set is labelled as MIAS_TS.

Table 5 Final feature set used in the experiment and their importance.

Rank	Feature	Importance	Rank	Feature	Importance	Rank	Feature	Importance	
1	Cast ADC Profit	0.372038	9	Genre Success Share	0.023006	17	Cast Top Profit	0.011053	
2	Budget Cont	0.167000	10	Cast Total Movies	0.020826	18	Season	0.011034	
3	Dir Total Gross	0.064884	11	Genre Revenue Share	0.018179	19	Cast Total Members	0.009404	
4	Cast Mean No Movies	0.029325	12	Cast ADC Freq	0.017742	20	Cast top avg profit	0.008937	
5	Dir Avg Gross	0.028661	13	AGE	0.014721	21	Genre Freq Share	0.008166	
6	Competition	0.024513	14	Genre ROI Share	0.013170	22	Plot topic	0.006682	
7	Genre Budget Share	0.024290	15	Cast Novelty	0.013090	23	Genre(avg)	0.006602	
8	Cast Total Rev	0.023006	16	Total Tenure	0.011701	24	Avg Annual Profit	0.005113	

Portions of this text were previously published as part of a preprint (Shahid & Islam, 2021). As normal K-fold cross validation is not suitable for TS based data therefore, a nested cross-validation technique called the forward chaining is used which also avoid bias due to over-fitting and under-fitting (Mun & Chong, 2018; Dora et al., 2018; Parvandeh et al., 2020). The forward chaining method (Nested-CV) is used to create six instances of feature set for training and testing of the models and the feature set is divided based on time. Table 3 shows the movie distribution for nested cross-validation data instances.

The machine learning models trained on the different instances of feature set include random forest, AdaBoost, Gradient Boost, Xgboost, MLP, K-NN and SVM. Based on the model comparison (Fig. 9A), Gradient Boost classifier performs well and it is selected for detailed analysis. Benchmarks work are reevaluated on the current dataset and class labels (Hit/Avg/Flop) for uniform comparison. Using Lash & Zhao (2016) features vector, and Gradient Boosting model shows 0.804 accuracy. The Gradient Boosting model is also trained on Gao et al. (2019) features vector and achieved an accuracy of 0.841. Using the features proposed by Ahmed, Waqas & Afzal (2020), 0.51 accuracy is achieved using Gradient Boosting model.

Figure 9 Proposed approach performance comparison with other benchmarks.

As shown in Fig. 9B the results obtained by applying the Gradient Boosting model on proposed features (MIAS_TS), the accuracy of 0.92, F1-score (macro) of 0.926 and AUC-score of 0.944 is observed that outclasses the previous works. The accuracy achieved using proposed features outperforms the Ahmed, Waqas & Afzal (2020) by a margin of 35.7%, Lash & Zhao (2016) by 12% and Gao et al. (2019) by 8.3%. As the pre-processing details of the previous research article along with the final movie count in the data-set are not available, we have obtained the results by training the corresponding models using the features mentioned in the respective research articles to ensure a fair comparison. Detailed results are presented in Table 6.

Portions of this text were previously published as part of a preprint (Shahid & Islam, 2021). The significance of individual features in accurately predicting the movie success is evaluated by dropping the selected features gradually. In Fig. 10, “ - ” mean minus or drop out, B means “Budget share”, S means “Success share”, R means “ROI share”, G means “Gross or Revenue share”, F mean “Freq share”, and Bd_C means “Budget contribution”. Figure 10A shows that proposed features are contributing in improving movie profitability prediction results as drop out results in decrease in accuracy. To see the newly proposed genre popularity feature individual importance, analysis is presented in Fig. 10B. It shows that the major impact is from the “Budget share”, and the least impact is from the “Success share”. This analysis shows the among newly proposed time series based features, genre popularity represented through “Budget Share” contributes the most to predict the movie success.

Apart from that, to find the importance of features that are used in the model training, information gain is also calculated using the Sklearn (Pedregosa et al., 2011) builtin parameter “feature importance”. It tells that which feature is more effective during the training process. The feature importance score is presented in Table 5. Based on the feature importance, it can be said that if the movie has a team that has success in the past, has higher chances of producing another successful movie. Additionally, three novel features (budget contribution, genre budget share, genre success share) are present in the top 10 features. This highlights the significance of considering the features based on temporal aspect of Genre Popularity. Genre as a categorical feature is in the 23rd position in the list, while the time series based genre popularity feature made to the second most important feature.

Impact of change in movie release time (RQ 3): The following results are obtained form experiments that are designed to evaluate the impact of change in release time of the movies. As it is unreasonable to suggest a suitable release time for a movie in the absence of ground truth, the aim of the experiments is only to observe the change in behavior of the prediction algorithms.

Table 6 Multi-class classification results comparison using Gradient Boost.

Performance metric	Benchmark	MIAS_TS	
	Lash & Zhao (2016)	Gao et al. (2019)	Ahmed, Waqas & Afzal (2020)		
Accuracy	0.804	0.841	0.567	0.924	
Precision	0.810	0.843	0.579	0.926	
Recall	0.805	0.841	0.567	0.927	
AUC	0.856	0.879	0.663	0.944	
F1-Score	0.810	0.843	0.567	0.926	

Figure 10 Performance analysis of proposed features using dropout technique.

Five different instances of dataset are created by altering movie release time. In a first dataset instance movie release time is set 1–3 months prior to actual release time, shown with “−3” on the x-axis in Fig. 11A. “+ 6” shows that release time of movies is set to 6 months is later than the actual release time. Similarly, “1 and 2 Year” represents the plots where 1 and 2 years (12 or 24 months) have been added in the release time of the movies. Y-axis shows the percentage of the movies that have changed their profitability class. In end, a movie release time is set to the time which has maximum genre popularity of each genre in a year and called it “Popular”.

Figure 11 Impact of release time w.r.t genre popularity on movie profitability.

When the synthetic release dates are inserted in the dataset, the forecasting models are retrained based on the five instances of dataset. Based on the new feature set data, the profitability class labels, i.e., (Hit, Avg, Flop) are also updated. The change in success class is evaluated in two ways, i.e., absolute deviation and month-based deviation as shown in Fig. 11A. Absolute deviation refers to the movies that have switched their success class due to change in release time when their absolute ROI is considered. The month-based deviation refers to the movies that have changed their success class based on monthly ROI values. Synthetic ground truth is generated based on monthly ROI values, assuming that if the movie belonging to a certain genre is released in particular month, then the profitability of that month is used to assign the profitability class. New class label is assigned using the monthly ROI values along with the movie individual contribution in that monthly ROI values as shown in Eq. (5).

Based on (popular) dataset instance, 62.18% movie’s profitability class changes from the actual profitability class and 71.14% of the movies changes their profitability with respect to monthly ROI. 65% of the movie’s profitability changes when seasonality based genre popularity dataset is created. In comparison with the movie actual success class, on average 62.84% of the movie’s success class changes, which shows that with the change in genre popularity, movie success will be affected. On average 72.6% of the movie’s profitability class changes in comparison with monthly ROI based profitability class.

Figure 11B is presented to identify the profitability class that is most affected by change in release time. The first stacked bar with label “Actual Class” shows the  percentage of movies in each class, i.e., “Flop, Hit, Avg” movies based on the true dataset instance. In a non-altered dataset state, 33.8% movies are “Flop,” 37.8% are “Average,” and 28.4% are “Hit” movies. This is followed by the five set of stacked bars corresponding to each dataset instance discussed earlier. Each individual bar in the group correspond to the number of movies that have switched their profitability class belonging to the label of the bar. In the “Popular” dataset instance, the “Flop” stacked bars tells that movies having flop class in previous state, changes into flop, average, and hit classes in new state after altering release time. It shows that 11.7% flop movies do not change the class and remain flop, 38.3% flop movies change to average movies, and 50% flop movies change to hit movies. Based on this plot, with the change in genre popularity based release time, flop movies can be made more profitable. Also previously average movies retain the state and change to another class is less as compared to flop and hit movies. So this experiment highlights the importance of genre popularity based release time on the profitability of the movie. It also highlights that chances of better performance on box office is high of a movie when its genre popularity is high.

Conclusion and Future Work

Portions of this text were previously published as part of a preprint (Shahid & Islam, 2021). To support a movie investment decision, this work highlights the importance of the temporal aspect of the genre. In this research, a time series based genre popularity features are proposed and defined in terms of revenue, budget, success, frequency, and ROI. Multiple machine learning models are evaluated on the MIAS_TS (proposed) feature set and best performing model is selected based on the high accuracy and F1-score values. Based on model selection experiment, Gradient Boosting classifier is selected, trained on feature set, and got the accuracy of 0.924, also F1-score of 0.926 and AUC-score of 0.944 is achieved. This work outperforms the Lash & Zhao (2016) by 12%, Gao et al. (2019) by 8.3%, and Ahmed, Waqas & Afzal (2020) by a margin of 35.7% in accuracy metric. Also to evaluate the magnitude of ROI, regression based analysis is performed and MLP regressor is selected from model comparison experiment which is further trained on the MIAS_TS dataset to predict the ROI values and got the MSE of 2.398 × 10−6. The success of movies is also dependent on previous collaboration of the team (actors and directors). The more experienced team and profitable previous collaborations are, higher the chances to achieve success as shown in Table 5 Cast-ADC-Profit is the most important feature.

This work shows that time series based genre popularity features promise a significant improvement in the results to predict movies’ success. In future, multivariate genre popularity time series forecasting will be explored. Also the concept of genre popularity will be applied to generic product popularity to see how much improvement this five dimensional product popularity feature will made in improving the business investment decisions.

Additional Information and Declarations

Competing Interests

Author Contributions

Data Availability

The authors declare there are no competing interests.

Muzammil Hussain Shahid conceived and designed the experiments, performed the experiments, analyzed the data, performed the computation work, prepared figures and/or tables, authored or reviewed drafts of the article, and approved the final draft.

Muhammad Arshad Islam conceived and designed the experiments, analyzed the data, authored or reviewed drafts of the article, and approved the final draft.

The following information was supplied regarding data availability:

The code used to perform the experiments is available at GitHub and Zenodo:

- https://github.com/muzammilhux/MIAS_TS-Paper

- Shahid, Muzammil Hussain, & Islam, Muhammad Arshad Islam. (2020). Code for Investigation of time series-based genre popularity features for box office success prediction. Zenodo. https://zenodo.org/records/8126162.

The dataset is available at Kaggle: https://www.kaggle.com/datasets/rounakbanik/the-movies-dataset.

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
