# Peer review of "Investigation of time series-based genre popularity features for box office success prediction"

_PeerJ Computer Science, doi:10.7717/peerj-cs.1603_

## Round 0.1 · original submission · Major Revisions

Dear Authors,

Please carefully revise the paper as per the suggestions of the reviewers, additionally, the following concerns should also be addressed by AE

1. you have mentioned in the abstract that you have used machine learning classifiers. Can you please explicitly elaborate/name which ones you have used?
2. Pl. justify how your solved problem is a multi-class problem.
3. In an abstract state the validity of your findings with respect to state-of-the-art work. eg. what validation techniques you have used? and how they outperformed the existing studies.
4. pl. redraw Figure 1 to make it easy for color-blind people.
5. few language problems should be fixed in revision

Reviewer 1 ·

Basic reporting

Overall paper is well written, clearly structured and presents key findings in clear manner. The authors provided sufficient background and context for the field, with an extensive review of relevant literature and references to prior research. However, almost all the references are published before 2021 and a few works have been used that consider early prediction of movies' success.

Methodology is thorough and transparent, outlining the data sources, and algorithms used in the study with the use of figures where appropriate.

Experimental design

The paper proposes a method to predict the success of movies at an early stage by analysing the changing behaviour of genre popularity. However, the authors do not provide sufficient evidence to support their claim that genre popularity does not follow a predictable pattern. Although some analysis is provided in Figure 6, it requires further elaboration, and it would benefit from being associated with the problem statement. The Y-axis of Figure 6(a) is ambiguous.
Fig 3 and Fig 4 suggest that a few genres always yield low ROI. What could be the motivation of investors to invest in such genres?
The plot shown in Fig 3 and Fig 5 are somewhat ambiguous. What does negative ROI represent in Fig 3?
Authors have provided criteria for identifying a movie as a hit or flop in Table 3. Authors should provide details on the methodology used to arrive at these numbers. How a movie with ROI less than 1 can be justified as average?
Figure 7 is not readable. Its size should be increased to understand it better.
Authors should identify the genre, which can be considered a good example for popularity forecasting.
Authors should include the number of movies considered in each time window of Fig 10 for clarity. How the results presented in Fig 10(b) have been obtained for previous work?
Which time window has been used to present the results in Table 6?

Validity of the findings

The paper presents an interesting and relevant topic, as predicting the profitability of movies at an early phase of production is a valuable tool for investment decision-making. The paper presents an interesting and relevant topic, as predicting the profitability of movies at an early phase of production is a valuable tool for investment decision-making.

However, there are some limitations to the study. The sample size used for experimentation is relatively small, and the study could benefit from a more extensive dataset. Additionally, the authors could have discussed its generalizability to other movie markets or production contexts.

Reviewer 2 ·

Basic reporting

The author claimed the novel “Genre popularity” features in terms of Budget, Revenue, Frequency, Success, and Return on Investment (ROI) for a successful investment in a movie production.
The author claims an improvement of 37 percent accuracy from existing approaches.

However, the literature seems too old, comparison approaches with proposed approach is form 2016 and 2020 (2 years back). Recent survey is required explore the validity/impact of features claimed by author.

Experimental design

The research question are relevant and valid with the topic of “Genre popularity” features using time series prediction.

Validity of the findings

The feature selection is different from existing approaches which concludes the novelty of the approach. The approach has a powerful benefit in investment prediction with newly selected features.

The major focus of proposed approach is on time series feature, the data set used for experiments is published in 2017.
However, experimentation of proposed approach on some recent time based movie dataset is required to validate the importance of time feature. It doesn't mean any objection on the proposed approach procedure.

Reviewer 3 ·

Basic reporting

'no comment'

Experimental design

'no comment'

Validity of the findings

'no comment'

Additional comments

In this manuscript, the main focus is to carry out time series-based prediction of movies revenue prediction. It is an important research study as proper prediction and recommendation of movies can yield a high increase in revenue in the movie industry. The main novelty of the research work is that it also considers the Genre related features as well. Overall, the research is presented in a proper manner, the flow of the content is good, the arguments are discussed in a logical manner. Although the level of the research work is satisfactory, the following minor comments may help the authors to improve the manuscript.

1) Table 1: If in TS column which shows time series data, all the values are NO then there is no need of this column. As it can be discussed in text that none of the existing studies lack to consider time series analysis. In addition, why three references in first column of the table are bold?
2) In “RQ 1: How accurately can we forecast “Genre popularity”?” it is recommended that instead of adding personal noun we, a general research question can be formulated, such as “RQ 1: How accurately “Genre popularity” can be predicted?
3) Figure 2: it needs improvement such as
a. Data preprocessing is carried out for data cleaning. While both steps are separately mentioned.
b. The term “reduce feature set” should be “reduced feature set”
c. Results should be shown after application of the proposed model of MIAS_TS model. While it is shown separately and linked with comparison. From model to results linkage is expected.
4) Algorithm 1: it is good that proper algorithm is shown. However, “for all” loop can be replaced with foreach loop. Which is proper as it shares element by element traversal in a data structure which is appliable in list of movies.
5) The manuscript is well written however it can be proofread for minor alterations. Such as line 271, movies released in ti should be number of movies released in ti 271 .
6) Line 378 states: “Synthetic time series are generated for comparison with the SARIMAX model’s forecasted value”. It shows that synthetic time series data has been used. While the authors have mentioned IMDB Data then why synthetic time series data has also been considered for comparative analysis?
7) Line 419 states: “features (MIAS TS), the accuracy of 0.92, F1-score(macro) of 0.926 and AUC-score of 0.944) is observed 420 that outclasses the previous works. “ why 0.92 is bold while others are not. I think bold in expected in table not in text.

Reviewer 4 ·

Basic reporting

The paper focuses on predicting the success of box-office movies by utilizing available features during the pre-production phase. Apart from considering features related to the movie itself, such as cast attributes and budget, the authors propose a novel prediction feature: genre popularity at the time of the planned movie release. This genre popularity is predicted based on historical time-series data. The authors demonstrate improved prediction results by incorporating these proposed features. Having said this improvements are suggested.

Experimental design

In Section 3.2 (page 8), the authors mention the selection of eight features that are available during the pre-production phase, including budget, genres, and revenue. However, it seems there may be an error regarding the inclusion of revenue as it is not known during the pre-production stage.

Regarding the non-obvious features, more information is requested about the "overview" feature and how it is coded. It would be helpful to provide additional details to clarify its representation and coding methodology.

In Section 4, where the authors mention splitting the feature data into training and test sets, more information is needed about the training procedure, including parameters and the specific methodology used for dataset splitting.

In Section 5 (page 16), there is a question regarding the date range "next 9 years (2010-2019)" and how it relates to the previously mentioned date range of 1972-2018. Clarification is required to understand the connection or distinction between these two date ranges.

In Table 1, it is suggested to include information about the dataset used, such as the name or source of the dataset and the number of movies included in the analysis.

For Figure 5a, a request is made for a more detailed description of the information represented on both axes to enhance understanding.

Regarding Figure 5b, clarification is needed regarding whether the values represented are averages or cumulative values

Validity of the findings

Section 5, which discusses the results, mistakenly presents the list of classifiers, which should actually be included in Section 4 on Experimental Details. It is crucial to provide information about the prevalence of each class in the dataset to assess whether it is imbalanced or not, as this can have implications for accuracy.

On Page 2, there is inconsistency in the naming conventions used for "Genre Popularity," including variations such as "Genre Popularity," "Genre popularity," and "Genre popularity" within a single column of the paper. This inconsistency should be addressed for clarity and coherence.

The placement of figures also appears to be haphazard, with instances where Figure 3 is displayed on Page 8 while the corresponding text on Page 8 references Figure 7. It is recommended to position figures as close as possible to the relevant text that references them to improve readability and understanding.

Additional comments

Specific comments regarding Figure 1: where the legend and axis labels are too small to read, and the genre order in the legend does not match the order in the stacks, making it difficult to interpret when printed in grayscale.

Overall, these issues make the paper challenging to read and comprehend. In its current state, it is not suitable for publication, and a major revision is strongly advised. The authors are encouraged to carefully proofread and reorganize the paper to address these concerns.

---

## Round 0.2 · accepted · Accept

Congratulations! The revised version of the paper is accepted based on my reading and the reviewer's recommendation.

Reviewer 4 ·

Basic reporting

After a thorough reassessment of the revised manuscript, it becomes apparent that the authors have diligently incorporated the feedback offered by both the reviewers and the editorial team. The paper has undergone substantial enhancements, notably through the incorporation of contemporary reference materials. These revisions have notably contributed to augmenting the lucidity and overall caliber of the content.

Experimental design

Correctly executed and appropriate to the domain

Validity of the findings

Correct and well discussed

Additional comments

All observations addressed. Accept!